# Stable, Low Power and Bit-Interleaving Aware SRAM Memory for Multi-Core Processing Elements

Nandakishor Yadav [1,*,†], Youngbae Kim [2], Shuai Li [2] and Kyuwon Ken Choi [2]

1 Fraunhofer Institute of Photonics Microsystems IPMS, 01109 Dresden, Germany
2 Illinois Institute of Technology, Chicago, IL 60616, USA; ykim102@hawk.iit.edu (Y.K.); sli97@hawk.iit.edu (S.L.); kchoi12@iit.edu (K.K.C.)
* Correspondence: nkyadav.vlsi@gmail.com
† The author executed this work when he was working at Illinois Institute of Technology Chicago, USA.

**Abstract:** The machine learning and convolutional neural network (CNN)-based intelligent artificial accelerator needs significant parallel data processing from the cache memory. The separate read port is mostly used to design built-in computational memory (CRAM) to reduce the data processing bottleneck. This memory uses multi-port reading and writing operations, which reduces stability and reliability. In this paper, we proposed a self-adaptive 12T SRAM cell to increase the read stability for multi-port operation. The self-adaptive technique increases stability and reliability. We increased the read stability by refreshing the storing node in the read mode of operation. The proposed technique also prevents the bit-interleaving problem. Further, we offered a butterfly-inspired SRAM bank to increase the performance and reduce the power dissipation. The proposed SRAM saves 12% more total power than the state-of-the-art 12T SRAM cell-based SRAM. We improve the write performance by 28.15% compared with the state-of-the-art 12T SRAM design. The total area overhead of the proposed architecture compared to the conventional 6T SRAM cell-based SRAM is only 1.9 times larger than the 6T SRAM cell.

**Keywords:** SRAM; stability; reliability; CNN; read time; write time





## 1. Introduction

Machine and deep learning techniques have been the main driving force of autonomous industries, such as aerospace and automobile industries, in recent years. These techniques are used for intelligent computation because of their major needs in autonomous industries. Autonomous robots are used to increase the productivity and reduce the cost of these industries. In the current technology, computation is performed by von Neumann architecture-based microprocessors [1]. This microprocessor architecture requires high-performance on-chip memory and an arithmetic logic unit. Therefore, improving the capabilities of hardware components, such as high-speed processors and significant space memories, are needed to provide more opportunities for high-performance deep learning architectures. The convolutional neural network (CNN)-based algorithm is the most popular deep learning algorithm [2,3]. Using shared weights that are automatically learned by the network, CNN is used to automatically determine the object features by implementing a convolution operation across deep layers of the network [4].

It is made of fully connected (FC) layers that form part of the CNN architecture, and is usually used for softmax classification and bounding box regression. FC layers usually contain a large number of neurons, which are fully connected with previous layers through weights. Figure 1 shows a single FC layer [3]. CNN needs parallel data processing for the computation and decision process, which is a process for CNN. It is carried out by multiplication and addition acronym (MAC) operations. ALU carries out this operation in the general purpose microprocessor. To improve the working efficiency, researchers have given a new hypothesis that is beyond the von Neumann architec-

ture. In this hypothesis, built-in computational memory is used to perform computation. This helps parallel computation for searching problems, such as on CNN. Some of the built-in computation memories are proposed by researchers, such as resistive random access memory (R-RAM) and static random access memory (SRAM) [3,5]. R-RAM cannot be fabricated by conventional CMOS technology. As a result, R-RAM may not be the preferable solution for on-chip memory. Instead, SRAM could be a preferable solution.

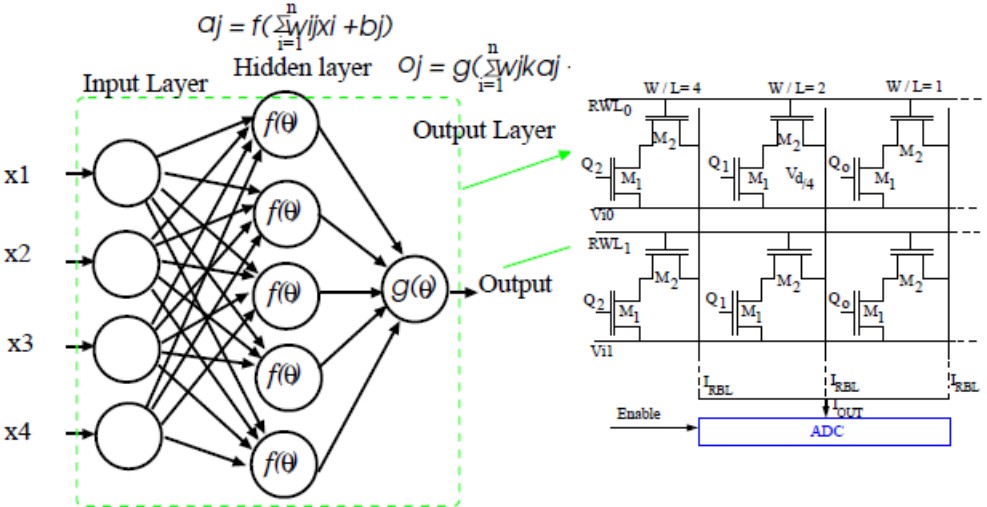

**Figure 1.** Basic CNN architecture and use of built-in computational memory.

Most successful built-in computing memory (BCM) is developed using non-volatile memories (NVM), such as resistive, spintronic, and ferroelectric memories. These memories use 1T, 1T1C, or 2T1C-based bit-cells [6,7]. An array of bit-cells is used to develop a sum-of-product or product-of-sum arithmetic. However, most of these NVM memories are not compatible with CMOS technology; hence, volatile memories (VM), such as SRAM and DRAM, can be used to develop BCM. Some of the existing SRAM-based computation memories are 6T and 8T SRAM cells. Don et al. [8] have presented Boolean vector operations using 6T SRAM cells. Besides, Jaiswal et al. [2] have demonstrated that the 8T cell can also be used for the sum-of-product (SOP) operation, as shown in Figure 1. The read port current provides the dot product, which senses the output current at the read port. They use a current-to-voltage converter followed by an analog-to-digital converter. Developing an arithmetic operation in the SRAM is not the only solution for a high-performance processor design for machine learning. The SRAM should also be stable and reliable. In addition, as most autonomous applications are remotely deployed, low power is also a major design challenge.

The SRAM should not lose the stored data in standby mode and read operation, and is given by the stability parameters, such as the static noise margin (SNM) and read noise margin (RNM) [9]. The process variation sources mostly impact the stability of the SRAM. Process variation sources exist, such as random dopant fluctuation and line edge roughness. The reliability of modern technology is reduced due to bias temperature instability (BTI) and hot carrier injection (HCI) [10]. A high stability of the SRAM cell can tolerate the process variation, BTI, and HCI effect. Hence, it requires extra transistors to increase the stability; however, these additional transistors sometimes reduces the performance and increase the power dissipation.

Supply voltage scaling is the most significant effective way to reduce both the switching power and leakage power for the VLSI circuit design [11]. However, designing resilient SRAMs for near-threshold or subthreshold operation is exceptionally demanding due to increased device variations and reduced design margins at low supply voltages with highly scaled processes. A 6T SRAM cell is the basic SRAM cell structure, but it has a read and write conflict and is most unstable at scaled supply voltages and nanoscale technologies.

However, the conventional 6T cell's minimum operation voltage is limited to around 700 mV, while logic circuits are designed to work at subthreshold operation. Therefore, a stable and high-performance SRAM is required for the subthreshold mode of operations [12].

SRAM cells are proposed by the researchers to improve the stability and performance. Read and write assist circuits are used to improve performance. After 6T, the 8T SRAM cell is the millstone technology. The 8T SRAM cell improves the read stability using the subthreshold read mode of operation [9,13]. In these schemes, cell storage nodes are decoupled from the bit-line during the read operation to increase the readability. Similarly, 9T, 10T, 11T, and 12T SRAM cells reduce the power dissipation [14–19]. These SRAM cells use supply feedback in order to internally weaken the pull-up current during write cycles and to improve the write performance of the SRAM cell. The write performance depends on the load transistor (should be weak) and access transistor (should be strong), whereas the read performance depends on the driver transistor (should be strong). Many assist circuits are proposed using the above-mentioned hypothesis, and the techniques are word-line boosting, multi-supply voltage, and negative bit-line voltage. However, the peripheral and assist circuits are capable of global and local variations. For example, negative bit-line (NBL) is an effective scheme to improve the write ability.

In the PPN 10T SRAM cell, positive feedback is introduced to increase the read stability, and a similar technique is used in the Schmitt trigger-based SRAM [19–22]. The read stability is increased in PPN 10T and Schmitt trigger-based SRAM, but the write performance is reduced. Furthermore, to improve the stability of the SRAM cell under process variation, BTI, and HCI, the 12T SRAM cell is proposed by [20]. It has two write word lines, one read word line, and a data-dependent write mode of operation. The two write word lines used to remove the bit-interleaving effect during read/write half select the cell. To remove the extra control signal and to improve stability using the recharge feedback circuit, we proposed a 12T SRAM cell. Recently, a 12T-based quadruple cross-coupled SRAM cell has been proposed for reliable applications [21]. It is a very complicated SRAM cell; hence, it requires three levels of materialization, which increases the area overhead. Gupta et al. [22] proposed a 12T SRAM cell to reduce the power dissipation. The modified cross-coupled inverter reduces the power many times more than the state-of-the-art design, but, at the same time, it reduces the stability and read performance. We proposed a bit-interleaving aware SRAM cell using the PMOS-based positive feedback, which further helps to increase the stability. Furthermore, the positive feedback works as refreshing logic during the read mode of operation. Further, we proposed a butterfly-finding SRAM architecture to reduce the power dissipation and increase the performance. The primary objectives of the proposed work are:

- We propose a bit-interleaving aware 12T SRAM cell;
- Positive feedback in introducing SRAM cell to increase the stability;
- Selective word line is proposed to increase the write performance;
- We propose a butterfly-based SRAM architecture to increase the performance and reduce the power dissipation;
- We compare the result with the state-of-the-art design.

The remainder of the proposed work is as follows. Section 2 discusses the proposed SRAM cells, read/write operation, stability, performance, and area analysis. In Section 3, the proposed SRAM architecture and control signals are discussed. Simulation results and the comparative analysis are discussed in Section 4. Finally, in Section 5, the conclusion is drawn.

## 2. Proposed SRAM Cell

In this paper, we have proposed a stable and reliable SRAM. Positive feedback and write word line (WWL) are used to improve the read stability of the SRAM. An extra write word line also adds a bit-interleaving aware feature. The circuit diagram of the proposed read-ability awareness 12T SRAM cell (RS12T) is shown in Figure 2. A cross-coupled inverter in the SRAM cell is the basic formation of transistors to store (latched) the single

bit information. The basic latch is made up of M1, M2, M3, and M4 transistors. M5-M6 transistors work as access transistors in the write mode of operation, and M7-M8 also work as access transistors in both read–write modes of operation. Series-connected access transistors M5 and M9 (M6 and M10) are controlled by the WWL and word line (WL) control signals. The other two PMOS, which are the M7 and M8 transistors, provide an adaptive feedback mechanism to the storing node Q and QB. These transistors provide access between bit-lines and storing nodes. This adaptive mechanism follows the latching state. In read mode, bit-line (BL), and bit-line-bar (BLB) pre-charges use the precharge circuit. After this, WL will activate, and, simultaneously, M7 or M8 will be "ON" according to the latched value (stored value). One of the PMOS transistors activates and connects one of the bit-line with one of the storing node. If the latching state is Q = 1, then M7 will be active and will refresh the Q node. In this way, the proposed mechanism increases the stability of the SRAM cell in the critical read mode of operation. The whole process takes a tiny amount of time due to the strong capacitive bit-lines. The M11 and M12 transistors provide a sub-threshold mode of the read operation. Control signal states in the write, read, and hold mode of operations are shown in Table 1.

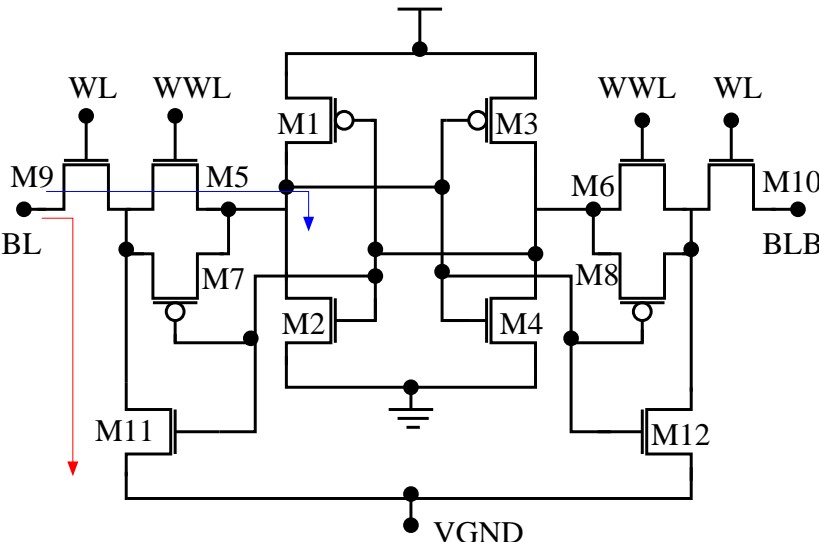

**Figure 2.** Proposed read stability aware 12T SRAM cell (RS12T).

**Table 1.** Control signals of the SRAM cell.

| WL | WWL | BL | BLB | Operation |
|----|-----|-----|-----|-----------|
| 0 | 0 | 1 | 1 | Hold |
| 0 | 1 | 1 | 1 | Hold |
| 1 | 0 | 1 | 1 | Read |
| 1 | 1 | 1/0 | 0/1 | Write |

In the write mode of operation, the WL and WWL control signal will be "ON", and single-bit data will write in storing nodes using series-connected M5, M6, M9, and M10 transistors. Two series-connected transistors, M5 and M9 (similarly, M6 and M10), increase the write path resistance and decrease the writing performance. This can be improved using strong access transistors or boosted WL and WWL signals. This technique requires an extra assist circuit, which increases the design cost. Another solution is to provide a single access transistor-based write operation, which is shown in Figure 3, i.e., the write operation can be carried out by BL and BLB transistors directly. It is a read stability and write performance aware 12T SRAM cell (RSWA12T), and increases the write performance and subsequently decreases the power dissipation. Furthermore, a read-stability-boosting

transistor can also be replaced by the NMOS transistors, as shown in Figure 4. However, this SRAM cell has some disadvantages, such as more NMOS transistors, as seen in the layout, which increases the well-proximity effect [23]. The proposed structure increases the read and standby stability. Furthermore, the proposed SRAM cell has a subthreshold read port through M9, M11, M10, and M12. The read operation does not modify the storing nodes (Q and QB). Hence, the read port can be used for computational memory design, as proposed in [2].

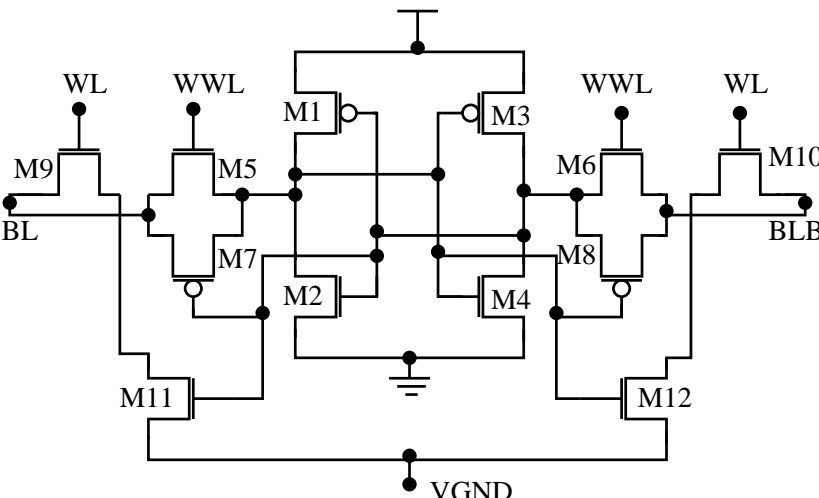

**Figure 3.** Proposed read stability and write performance aware 12T SRAM cell (RSWA12T).

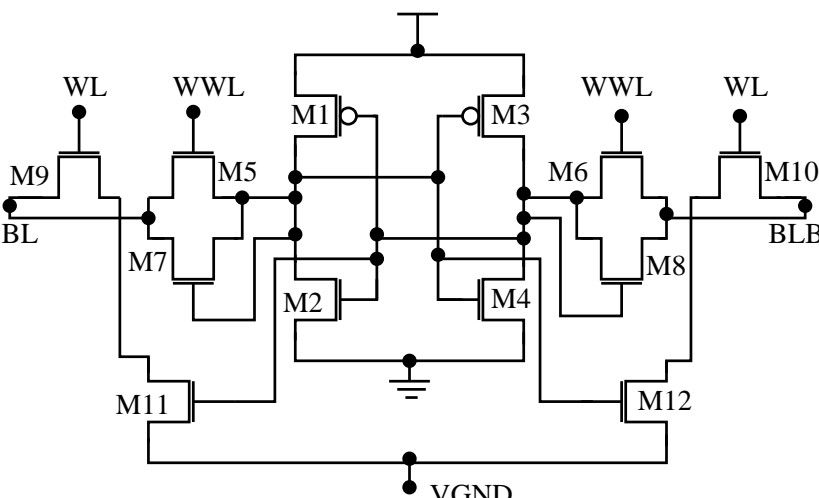

**Figure 4.** Proposed read stability and write performance aware 12T SRAM cell using NMOS transistors (RSWA12T1).

Furthermore, we analyze the static noise margin (SNM) and read the noise margin(RNM). We compare the proposed 12T SRAM cell with the existing 12T SRAM cell [20] and the conventional 6T SRAM cell. The internal structure of the proposed and existing SRAM cells is the same; hence, they have to have a similar SNM. In this work, we have improved the RNM. We have compared the simulation result using 45 nm freePDK CMOS technology [24].

Figure 5 compares the RNM for RSWA12T, exiting 12T, and conventional 6T SRAM cells. The results show a greater improvement in the RNM for RSWA12T than for the state-of-the-art 12T SRAM cell. In the proposed design, we also increase the write performance by a greater amount than the 6T and state-of-the-art 12T SRAM cell. The write trip margin or word line write margin (WLWM) is a wildly acceptable write performance parameter for the SRAM cell. Figure 6 shows the comparative WLWL results, which show a small increment in the write

performance for the proposed SRAM cell. We achieve this improvement because the proposed SRAM does not require the cell ratio; hence, we can increase the WLWM.

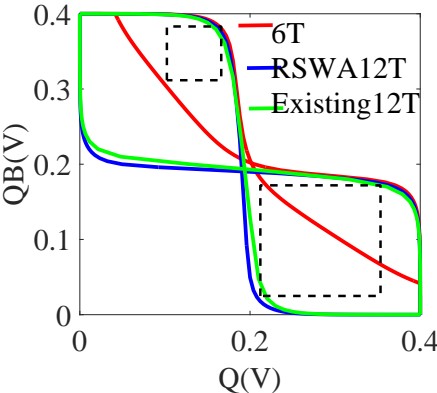

**Figure 5.** Comparison of RNM with conventional 6T and state-of-the-art 12T SRAM cell.

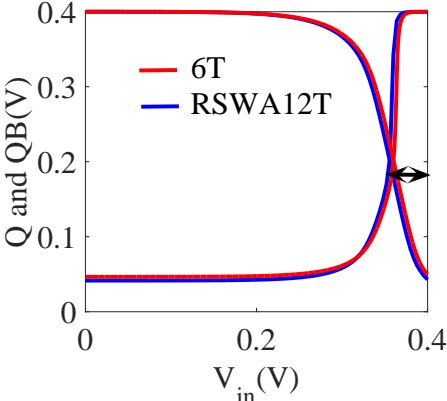

**Figure 6.** Comparison of WLWM with conventional 6TSRAM cell.

### 2.1. Read and Write Operation

In order to perform the read and write operation of the proposed SRAM cell, we have developed a 128-bit length bit-cell array, and the same is used for the 64 × 512 byte SRAM bank, where 64 is the data width, and the number of words is 512. The block diagram, with all of the control signals of the SRAM array, is shown in Figure 7. It consists of a PMOS-based pre-charge circuit, 128 SRAM cells, a word selector multiplexer, a read–write driver, and a sense amplifier. The read–write-driven designs use an inverter-based write driver [25]. In this design, the inverter is used to push and pull the voltage from the bit-lines (BL and BLB). The multiplexer is designed by using transmission gate (TG) logic [25]. TG-logic provides a better performance and increases the strength of signals. It is also a low-power technology; hence, it helps in saving the total power budget.

The latched type current sense amplifier is designed using ten transistors [26]. We have chosen a current sense amplifier because it gives a good performance. It also does not require critical tuning, and has a low offset. The current sense amplifier is also less affected by the offset voltage.

We analyzed the read and write performance when the read and write operations were at a 5 V supply voltage. The read and write performance for the proposed RSWA12T SRAM cell is shown in Figure 8. The simulation waveform shows the read and write operations for the sub-threshold mode, but, at this voltage, the read operation slows down. The read performance can be increased by using a high voltage. Figure 9 shows the read–write operation for the existing 12T SRAM cell. The simulation results show a greater improvement in the read and write performance for the proposed SRAM cell than the

existing 12T SRAM cell. We improve the write performance using the virtual ground signal (VGND). It is high during the write mode of operation and low in the read mode of operation, which helps to increase the write and read performance, respectively.

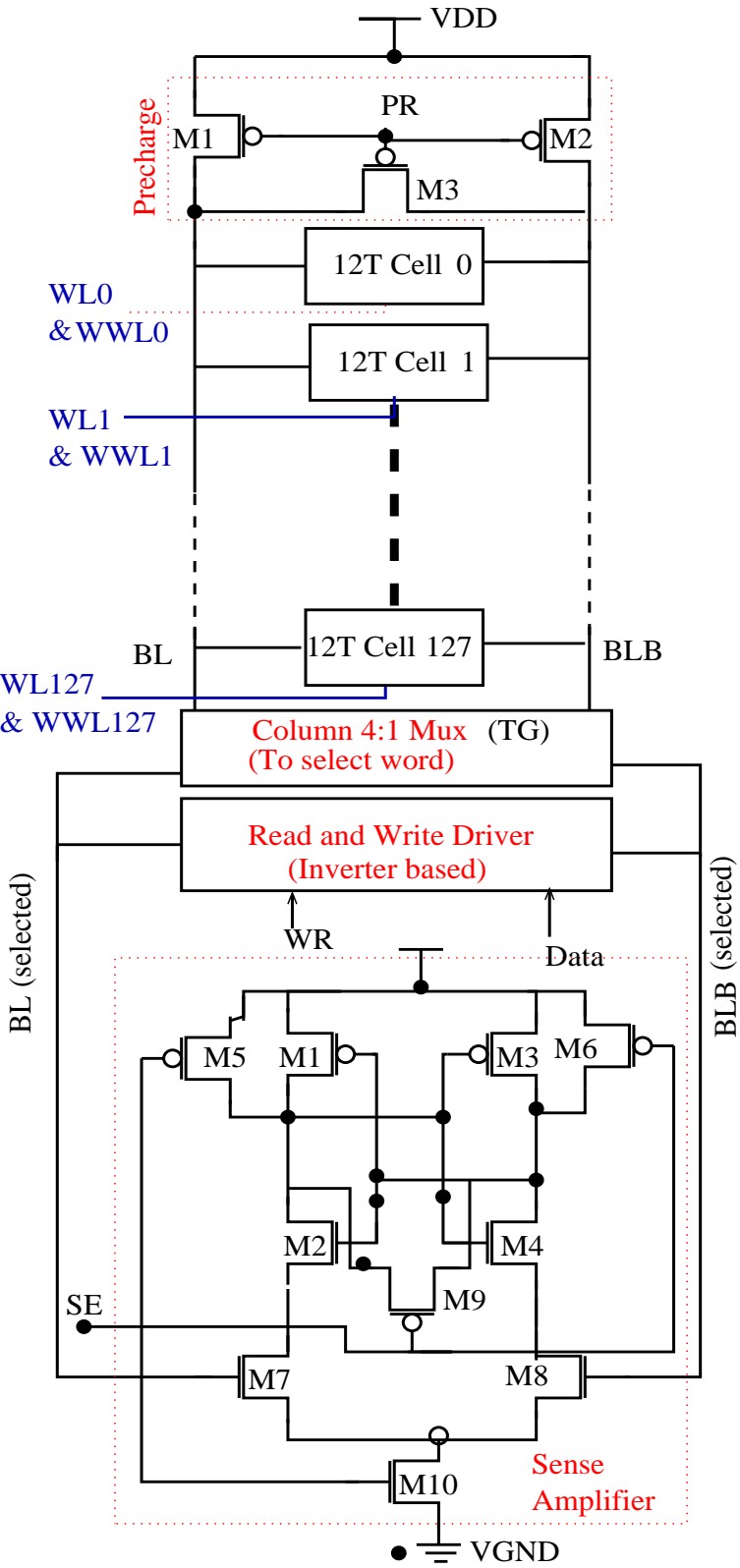

**Figure 7.** Proposed SRAM array and peripheral circuit, such as precharge, read–write drive, and sense amplifier.

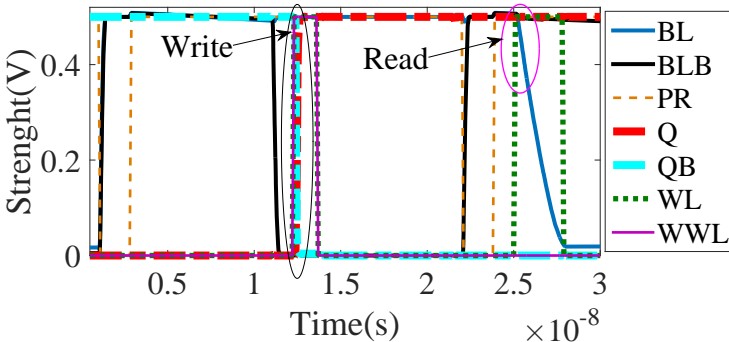

**Figure 8.** Read and write waveform and important signals for proposed second 12T SRAM cell at 0.5 V supply voltage.

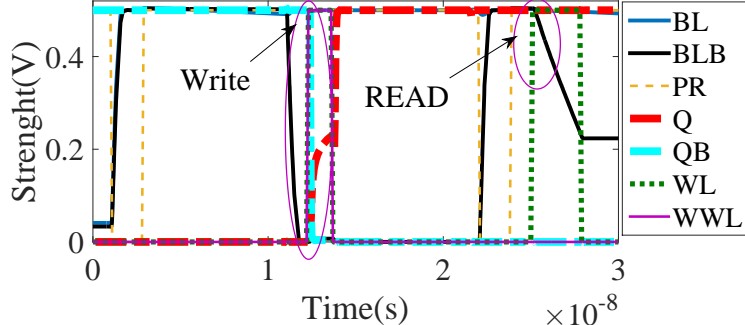

**Figure 9.** Read and write waveform and important signals for existing 12T SRAM cell at 0.5 V supply voltage.

Table 2 shows the comparative result for proposed and state-of-the-art SRAM cell-based SRAM arrays. The read time for the 0.4 V supply of the first proposed cell is faster than other cells, but this is a very small supply. The write performance is inferior to this cell: it is increased by 28.15% in the RSWA12T cell and 30.58% in the third cell, as shown in Figure 10. The comparisons for different supply voltages show that the read and write performance show a significantly negative change in the read performance and a positive shift in the write performance, which means that it follows conventional physics, such as in the 6T SRAM cell.

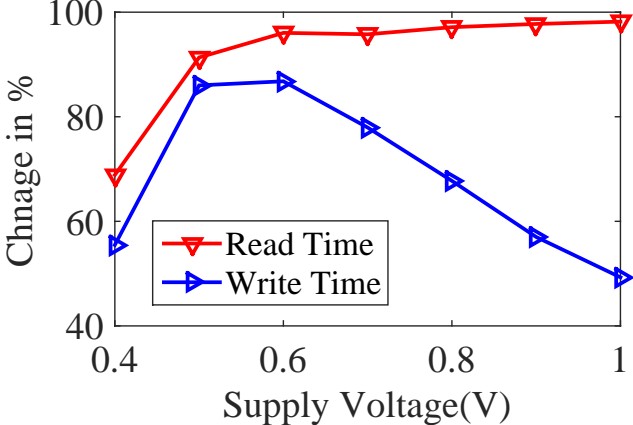

**Figure 10.** % Improvement in read and write performance in proposed RSAWA12 compared with the state-of-the-art design [20].

**Table 2.** Comparative performance analysis for 128 bit length array.

| Supply | RS12T | | RSWA12T | | RSWA12T2 | | Existing 12T [20] | |
|---|---|---|---|---|---|---|---|---|
| $V_{DD}$ | Read Time (s) | Write Time (s) | Read Time (s) | Write Time (s) | Read Time (s) | Write Time (s) | Read Time (s) | Write Time (s) |
| 1 | $7.61 \times 10^{-11}$ | $7.39 \times 10^{-11}$ | $1.26 \times 10^{-10}$ | $4.67 \times 10^{-11}$ | $1.21 \times 10^{-10}$ | $4.49 \times 10^{-11}$ | $6.97 \times 10^{-9}$ | $9.21 \times 10^{-11}$ |
| 0.9 | $9.79 \times 10^{-11}$ | $8.72 \times 10^{-11}$ | $1.57 \times 10^{-10}$ | $5.46 \times 10^{-11}$ | $1.59 \times 10^{-10}$ | $5.26 \times 10^{-11}$ | $6.97 \times 10^{-9}$ | $1.27 \times 10^{-10}$ |
| 0.8 | $1.21 \times 10^{-10}$ | $1.05 \times 10^{-10}$ | $2.01 \times 10^{-10}$ | $6.71 \times 10^{-11}$ | $2.00 \times 10^{-10}$ | $6.59 \times 10^{-11}$ | $7.00 \times 10^{-9}$ | $2.08 \times 10^{-10}$ |
| 0.7 | $1.81 \times 10^{-10}$ | $1.37 \times 10^{-10}$ | $2.97 \times 10^{-10}$ | $8.67 \times 10^{-11}$ | $2.93 \times 10^{-10}$ | $8.14 \times 10^{-11}$ | $7.02 \times 10^{-9}$ | $3.93 \times 10^{-10}$ |
| 0.6 | $1.93 \times 10^{-10}$ | $1.98 \times 10^{-10}$ | $2.78 \times 10^{-10}$ | $1.22 \times 10^{-10}$ | $2.84 \times 10^{-10}$ | $1.18 \times 10^{-10}$ | $7.02 \times 10^{-9}$ | $9.23 \times 10^{-10}$ |
| 0.5 | $3.91 \times 10^{-10}$ | $3.67 \times 10^{-10}$ | $6.10 \times 10^{-10}$ | $2.17 \times 10^{-10}$ | $6.10 \times 10^{-10}$ | $2.13 \times 10^{-10}$ | $7.04 \times 10^{-9}$ | $1.55 \times 10^{-9}$ |
| 0.4 | $1.39 \times 10^{-9}$ | $1.15 \times 10^{-9}$ | $2.21 \times 10^{-9}$ | $8.28 \times 10^{-10}$ | $2.07 \times 10^{-9}$ | $8.00 \times 10^{-10}$ | $7.10 \times 10^{-9}$ | $1.86 \times 10^{-9}$ |

*2.2. Bit-Interleaving Analysis*

We proposed a new SRAM architecture to reduce the bit-interleaving problem. Figure 11 shows the proposed SRAM architectures of selected, half-selected, and unselected cells, in which, Q and QB are the storing nodes. In order to achieve the six-sigma ($6\sigma$) yield requirements, all of the SRAM cells, including the selected cells (SLC), the row half-selected cells (HSLC1), the column half-selected cells (HSLC2), and the unselected cells (USLC), should tolerate a $6\sigma$ variation in their modes of operation. We carried out the routing direction of WWL in the RSWA12T SRAM cell in vertical. The way of simulation has already been discussed by Kang et al. [27]. The SLC cell of RSWAL12T arrays experiences the write operation and reaches $6\sigma$ yield expectations in the write mode of operation. Figure 12 shows the Monte Carlo simulation results of the write-selected cell, write half-selected cells, and unselected cells. The write time margin is very large, even at the random local variation of 10%. This shows that the bit-cell can work appropriately by designing a proper word-line pulse width between the write time. The proposed SRAM cell shows better results for bit-interleaving compared to the state-of-the-art 12T SRAM cell [21,22] and results that are similar to the [20].

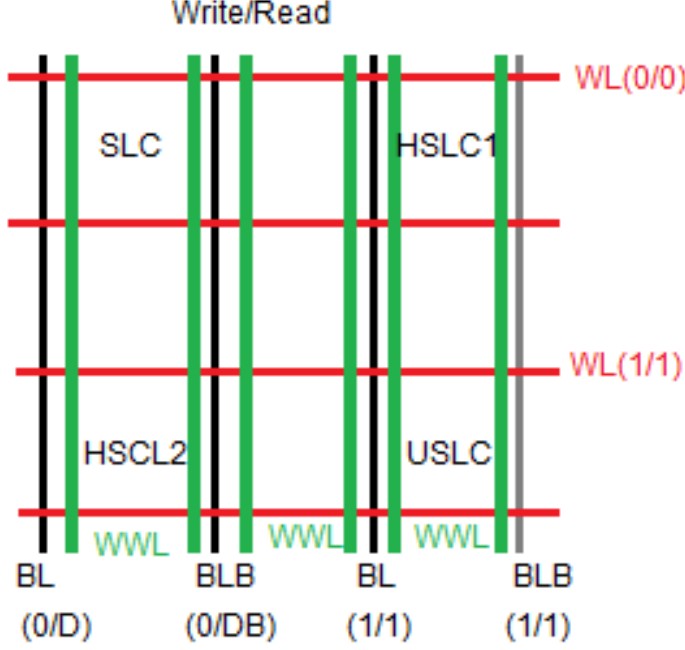

**Figure 11.** Bit-interleaving aware array architecture.

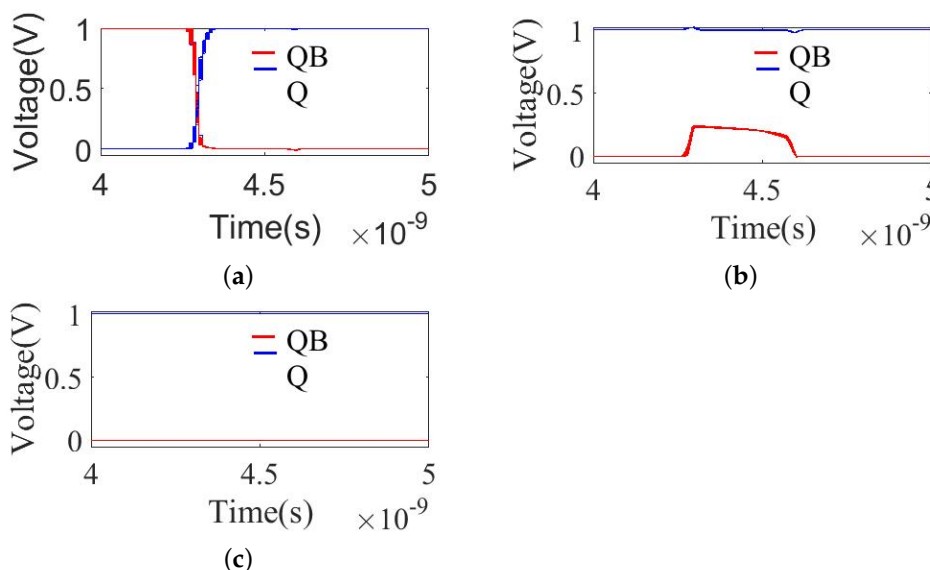

(a)

(b)

(c)

**Figure 12.** Simulated RSWA12T SRAM cell for write operation: (**a**) selected cell; (**b**) column half-selected cell; (**c**) row half-selected cell, for 5000 samples' Monte Carlo simulation.

## 2.3. SRAM Cell Area Analysis

In this section, we perform an area analysis for all of the proposed SRAM cells, and compare the results with the 6T SRAM cell layout. We use twin well-based 45 nm free PDK technology for the area analysis. We follow the ISO area for the performance and stability analysis, but we follow the standard sizing for the area analysis. The layout of the proposed RS12T cell is shown in Figure 13. We draw the N-well at the top and P-well at the bottom. Two access transistors are drawn horizontally, and the rest of the NMOS transistors are drawn vertically. We draw a compact layout to save the area. We draw well for the SRAM array so that we can share well for the compact SRAM array. The layouts for RSWA12T and RSWA12T2 are shown in Figures 14 and 15, respectively. Figure 16 shows the layout for the conventional 6T SRAM cell. The total area for all of the SRAM cells is mention in the side of the layout. The result shows that the proposed SRAM cell requires a 1.9 times larger area overhead than the 6T SRAM cell. In the next section, we shall discuss the SRAM architecture and total area analysis.

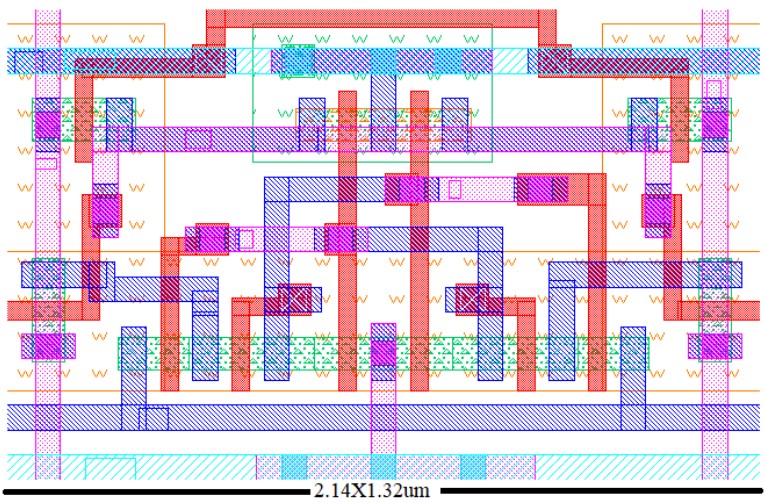

**Figure 13.** Proposed RS12T SRAM cell layout.

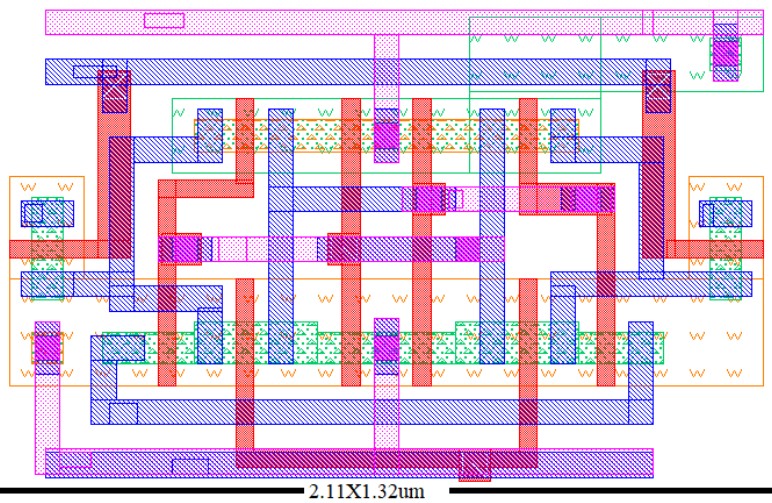

**Figure 14.** Proposed RSWA12T SRAM cell layout.

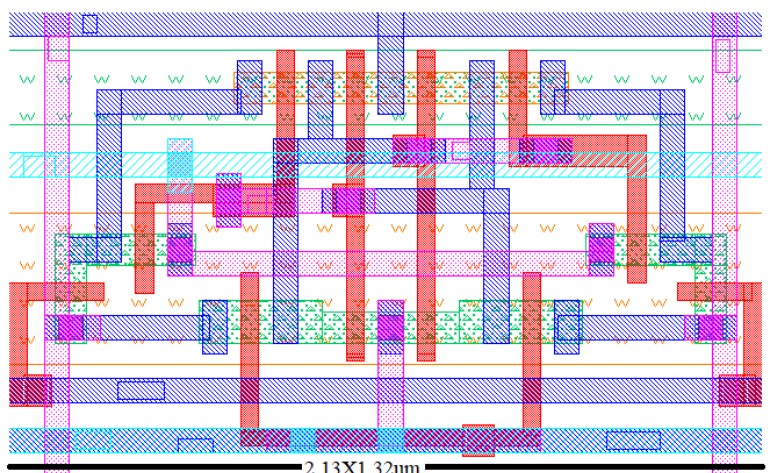

**Figure 15.** Proposed RSWT12T1 SRAM cell layout.

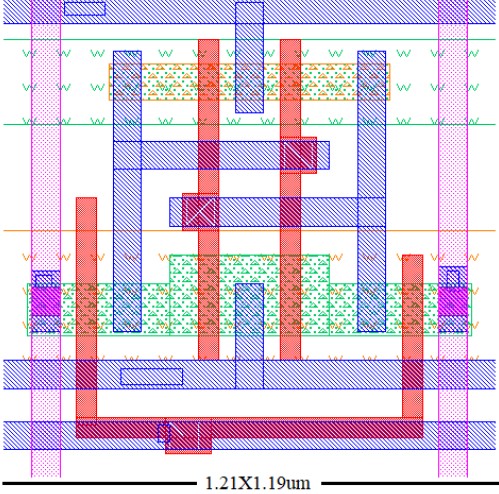

**Figure 16.** 6T SRAM cell layout according to the required sizing ratio.

## 3. Proposed SRAM Architecture

In this section, we discuss the proposed SRAM bank architecture and area analysis. We offer a butterfly-inspired SRAM architecture to save the power dissipation and to improve the performance. The block-level design for the proposed SRAM bank is shown in Figure 17. We used a 7:128 row decoder to select both sides of the SRAM arrays. AND arrays are used for the select logic for the WL and WWL signals. We have used two AND gate arrays for the WL and WWL signals, respectively. A 2:4 multiplexer (MUX) is used for the row decoder; we have used $464 \times 2$ MUX for 64 wide words and 64 read/write drivers and sense amplifiers, respectively. The total size of one SRAM bank is 512 bytes. Hence, the 1000 byte SRAM can be developed using two banks.

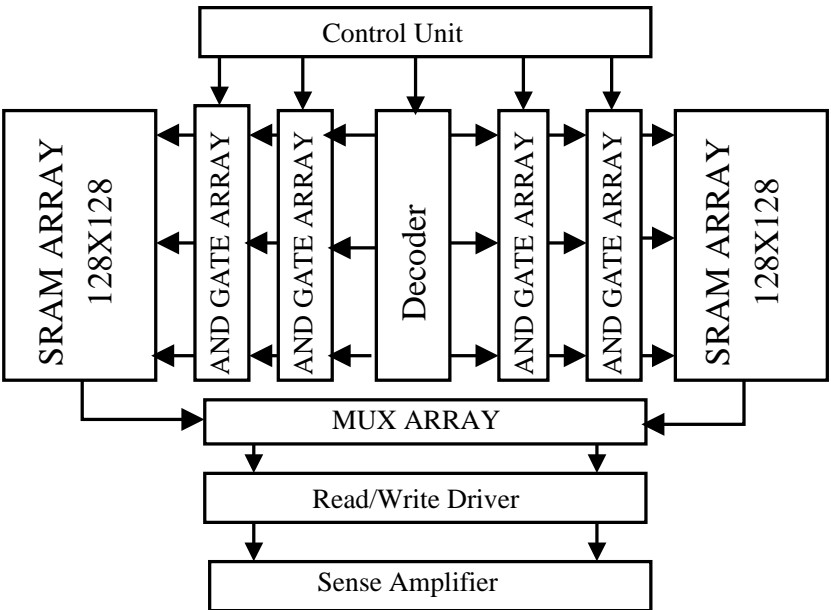

**Figure 17.** Butterfly-inspired SRAM architecture for low-power and high-performance SRAM bank.

The layout of the proposed SRAM without the I/O pad is shown in Figure 18. The total area of the 512 byte SRAM bank is $576.3 \times 170.1$ μm. The area occupied by the decoder, sense amplifier, driver, and multiplexer is only 6% compared to the SRAM array. The proposed butterfly-based layout also helps to reduce the total power dissipation and increases the performance. In this architecture, the row decoder selects only 32 columns from each side; hence, the data travel length is 32-bit only for the 64-bit output; hence it increases the performance. Furthermore, the buffer and divided word lines are used after the 32-bit width at each side in the butterfly architecture in order to increase the performance and reduce the power dissipation. A divided bit-line is also used for the low-power design.

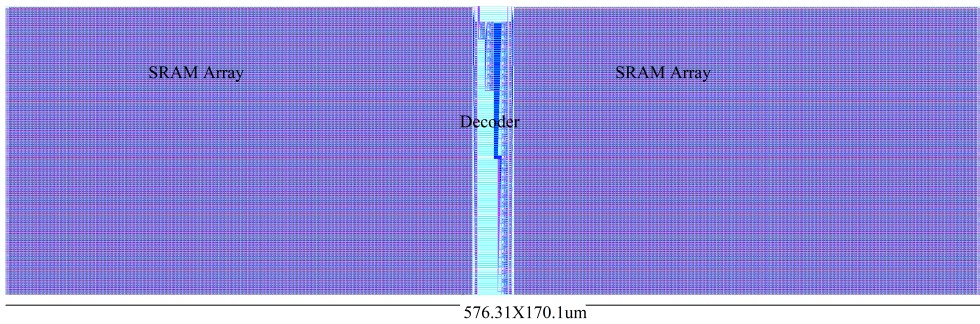

**Figure 18.** Layout of the proposed 512 byte SRAM bank: it has a decoder, SRAM array, sense amplifiers, multiplexers, and read and write drivers.

## 4. Simulation Result and Analysis

In this section, we will discuss the stability, performance, and power dissipation analysis for different supply voltages.

### 4.1. Stability Analysis

The stability of the SRAM is three types according to the three operating conditions. We discuss the standby, read, and write data margin by the static noise margin, read noise margin, and write noise margin. Furthermore, we compare the results with the conventional 6T SRAM cell and state-of-the-art 12T SRAM cells. We used the butterfly method for the stability analysis for the proposed and state-of-the-art SRAM.

SNM curves in the standby mode of operation for the proposed SRAM cells are shown in Figure 19. The result shows the stability results at different supply voltages. The SNM is the same for all proposed SRAM cells because the SNM depends on the latching inverters, which is the same for all SRAM cells. In the proposed work, we have increased the RNM using the extra transistors M7 and M8 in such a manner that these transistors help to increase the RNM. A similar RNM simulation result is shown in Table 3. The maximum SNM is provided by the RSWA12T cell. The butterfly curves for the same cell for different supply voltages are shown in Figure 20. The result shows a 2% and 49.9% greater increment in the RNM than the existing 12T SRAM cell and 6T SRAM cell, respectively, as shown in Figure 21. We increase the RNM through the self-adaptive M7 and M8 transistors. Transistors work as refreshing logic, such as dynamic RAM. In a practical case, the proposed SRAM cell will show a better RNM than the simulated results because of the feedback mechanism. Additionally, we used a subthreshold read mode using M11 and M12 transistors, and the transistors further increase the RNM.

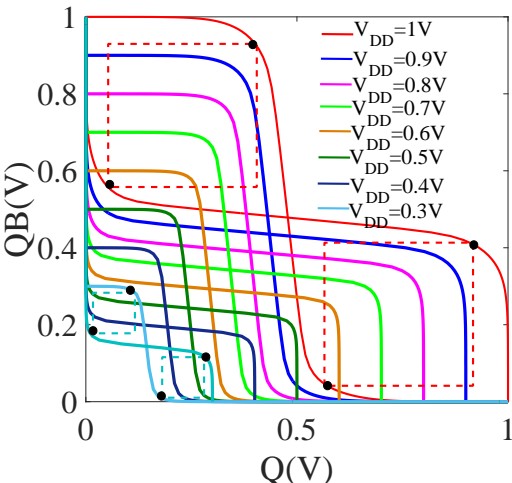

**Figure 19.** SNM butterfly curve for the proposed SRAM cell.

**Table 3.** Comparative stability results and analysis.

| | Proposed SRAM | | | | | | | | Existing | | | | | |
|---|---|---|---|---|---|---|---|---|---|---|---|---|---|---|
| | RS12T | | RSWA12T | | RSWA12T2 | | 6T | | Existing 12T [20] | | Existing 12T [21] | | Existing 12T [22] | |
| $V_{DD}$ | RNM(V) | SNM(V) | RNM(V) | SNM(V) | RNM(V) | SNM(V) | RNM(V) | SNM(V) | RNM(V) | SNM(V) | RNM(V) | SNM(V) | RNM(V) | SNM(V) |
| 1.0 | 0.3434 | 0.3470 | 0.3178 | 0.3470 | 0.3178 | 0.3470 | 0.1875 | 0.3617 | 0.3371 | 0.3371 | 0.1895 | 0.3354 | | |
| 0.9 | 0.3232 | 0.3300 | 0.3041 | 0.3300 | 0.3041 | 0.3300 | 0.1787 | 0.3428 | 0.3235 | 0.3235 | - | - | - | - |
| 0.8 | 0.2990 | 0.3050 | 0.2819 | 0.3050 | 0.2819 | 0.3050 | 0.1655 | 0.3151 | 0.3006 | 0.3006 | - | - | - | - |
| 0.7 | 0.2694 | 0.2721 | 0.2548 | 0.2721 | 0.2548 | 0.2721 | 0.1476 | 0.2794 | 0.2716 | 0.2716 | - | - | - | - |
| 0.6 | 0.2284 | 0.2344 | 0.2187 | 0.2344 | 0.2187 | 0.2344 | 0.1263 | 0.2397 | 0.2341 | 0.2341 | - | - | - | - |
| 0.5 | 0.1894 | 0.1903 | 0.1795 | 0.1903 | 0.1795 | 0.1903 | 0.1031 | 0.1974 | 0.1937 | 0.1937 | - | - | - | - |
| 0.4 | 0.1463 | 0.1487 | 0.1383 | 0.1487 | 0.1383 | 0.1487 | 0.0784 | 0.1524 | 0.1498 | 0.1498 | - | - | 0.1700 | 0.0544 |
| 0.3 | 0.0999 | 0.1051 | 0.0920 | 0.1051 | 0.0920 | 0.1051 | 0.0527 | 0.1061 | 0.1042 | 0.1044 | - | - | - | - |

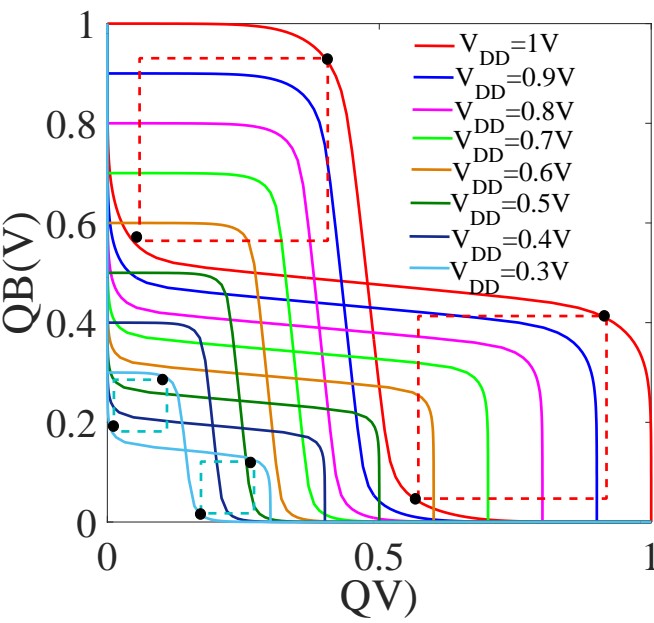

**Figure 20.** RNM butterfly curve for the proposed SRAM cell.

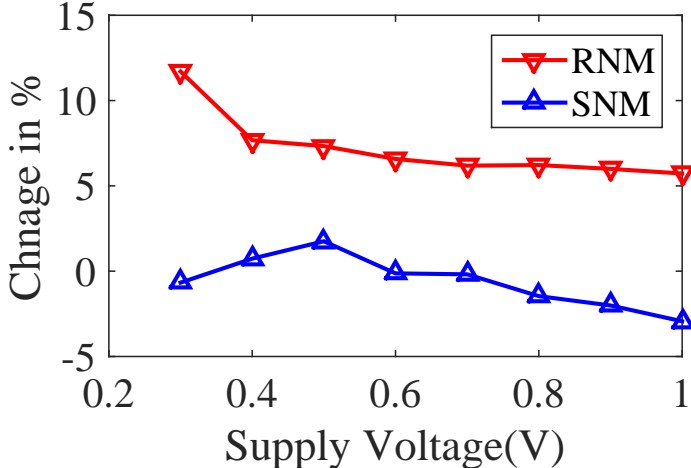

**Figure 21.** % change in the SNM and RNM of RSWA12T SRAM cell compared with state-of -the-art SRAM cell.

The reliability of the SRAM is reduced due to the process variation (PV). We further perform the stability analysis for PV. Figure 22 shows the PV analysis on the SNM for the proposed SRAM cells, and the result does not show a greater deviation due to PV. We increase the RNM and write performance; therefore, we analyze the RNM and WLWM for PV. The change in RNM due to PV for 5000 Monte Carlo simulations is shown in Figure 23. The proposed RS12T and RSWA12T SRAM cells are better than the RSWA12T2 SRAM cell. The deviation is also less than that of other cells, which shows less PV on the first and second SRAM cells.

The most stable cells out of the proposed cells are RS12T and RSWA12T, and the RSWA12T cell is proposed in order to increase the write performance. The WM butterfly curves for the second cell are shown in Figure 24. The WM is improved by 10% compared with the existing state-of-the-art 12T SRAM cell.

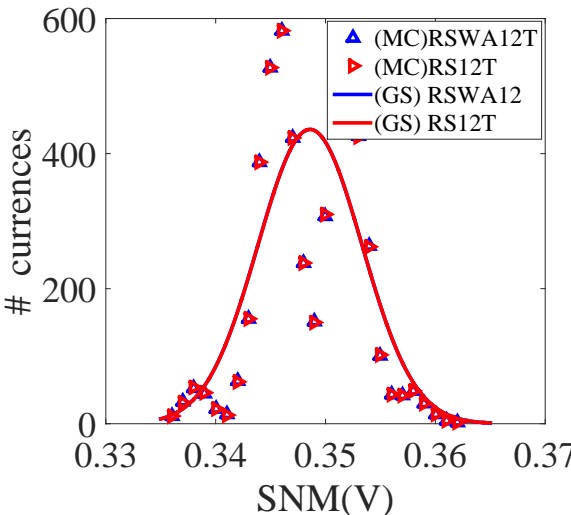

**Figure 22.** Process variation analysis of SNM for proposed SRAM cell.

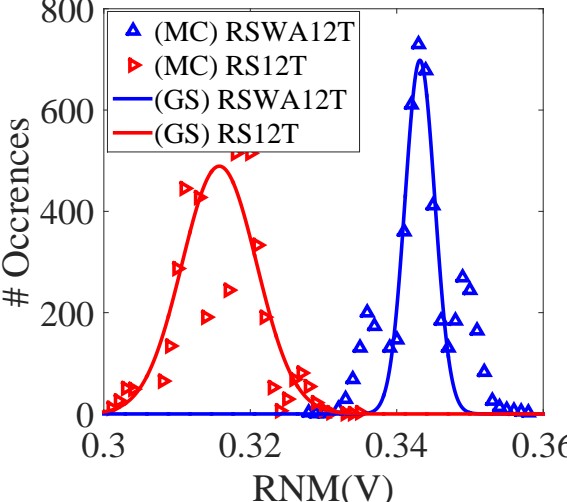

**Figure 23.** Process variation analysis of RNM for proposed SRAM cell.

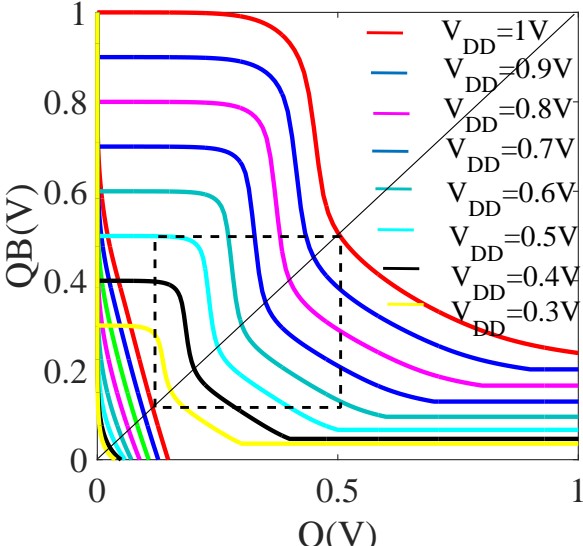

**Figure 24.** Write stability analysis for the proposed SRAM cell.

## 4.2. Performance Analysis

The writing performance of the SRAM cells can be analyzed using the trip point voltage at the SRAM cell level analysis. This is also known as a word line write margin (WLWM). In the proposed SRAM, we increase the RNM using M7 and M8 transistors. These transistors may decrease the write performance. However, we manage the write performance in RSWA12T and RSWA12T2 SRAM cells. A direct bit-line connection is provided for the M5 and M6 transistors, which helps to increase the write performance. Figure 25 shows WLWM for the first proposed SRAM cell; the result shows less WLWM due to series-connected transistors. The Write time is $n$ = boosted in the RSAWA12T. In this SRAM cell, we use the direct connection between bit-lines and access transistors. Figure 26 shows the WLWM results for the RSWA12T SRAM cell for different supply voltages. The WLWM is increased with a reduction in the supply voltage, as per the conventional physics for the SRAM. The simulation results for the reduced supply voltage show that the proposed SRAM can work in the subthreshold mode of operation.

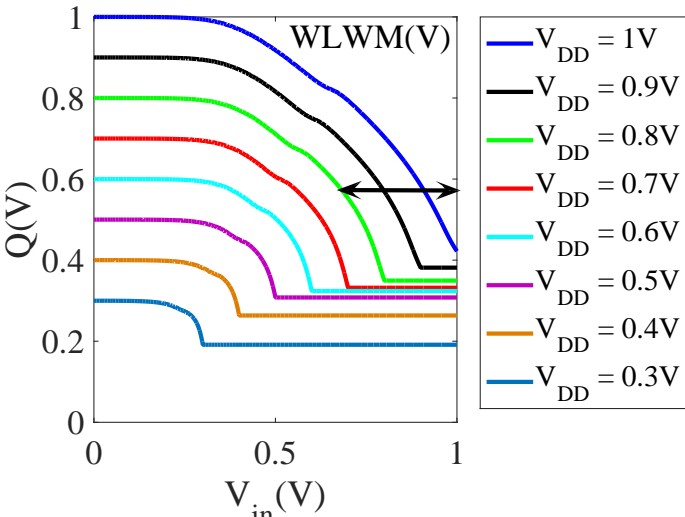

**Figure 25.** Word line write margin for first proposed SRAM cell at different supply voltages.

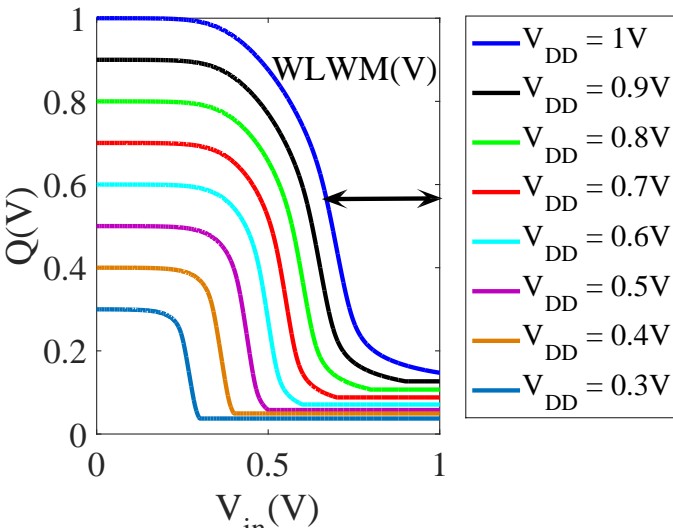

**Figure 26.** Word line write margin for second proposed SRAM cell at different supply voltages.

Furthermore, we carry out the change in WLWM under parametric variation due to process variation. We perform process variation analysis for typical globule process corners and a 1 V supply voltage. Monte Carlo distribution curves for all proposed SRAM cells are shown in Figure 27. The Monte Carlo and Gaussian equivalent curves show the mean ($\mu$) value and data distribution ($\delta$) for WLWM. The result shows much less distribution for the RSWA12T than for other SRAM cells. The result also shows an increment in the mean value; hence, RSWA12T is the best choice for further SRAM bank designs.

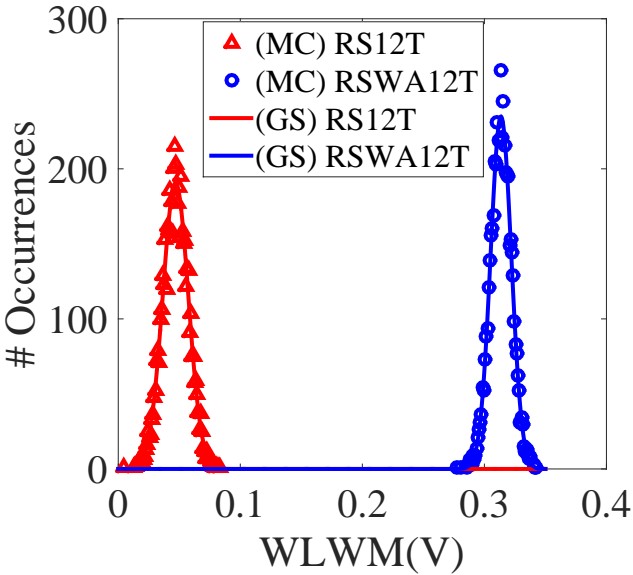

**Figure 27.** Process Variation effect analysis on proposed SRAM cells.

We have analyzed the read and write performance of the proposed SRAM bank and compared the result with the state-of-the-art design. The memory bank size is $128 \times 256$ bits, and the data width is 64 bits. Hence, the total memory bank size is 512 bytes. To design a 1000 byte SRAM, it requires two SRAM banks. We design a small SRAM bank to reduce the power dissipation. A memory controller is used to control the SRAM bank for a low power mode of operation. Furthermore, the proposed butterfly-inspired SRAM structure decreases the power dissipation. For the single SRAM bank, the estimated bit-line capacitance will be $C_{gs} \times 128$ and the word line capacitance will be $(C_{gs} + C_{gd}) \times 256$. In the proposed architecture, we have reduced the word line capacitance by half using the divided word line technique, which helps to increase the read and write performance. Furthermore, delay buffers are used to increase the signal strength and performance.

The status of the control signals and storing nodes for the RSWA12T SRAM cell for a convention structure and butterfly structure is shown in Figure 8. We only demonstrated important signals in the figure. The different control signals are a 64-bit input data signal, 7-bit input row-decoder, 2-bit input column-decoder, and 64-bit output data, etc. The read and write performance of the SRAM bank is shown in Figure 28. We simulated the SRAM bank for different supply voltages and checked the functionality when working for the low supply voltage. The simulation results show the time taken to write data into the SRAM cell when WL is activated, and the time taken to read data from the SRAM cell. The read time includes taking the time to read data from the SRAM cell and delivering it at the output port. The read and write performance is improved with an increase in the supply voltage. The state-of-the-art SRAM bank [21] was $123 \times 128$ bit and designed at a 130 nm technology node. Under ISO conditions, we improved the write performance by 15% and write performance by 8.6%. The performance also increased with the signal strengthening buffers. We used three buffers in the word line after every 64 columns.

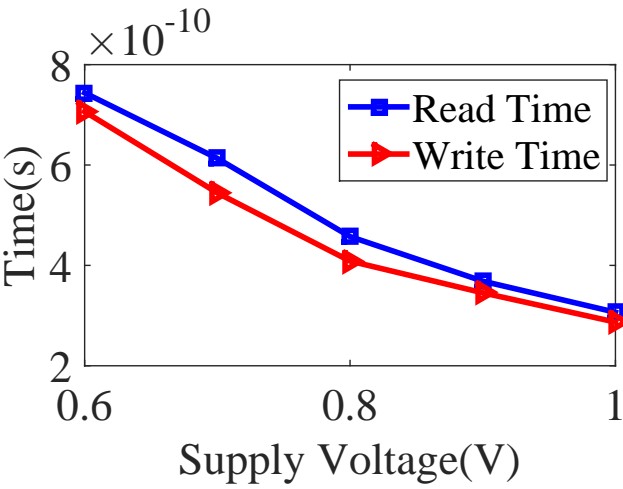

**Figure 28.** Simulation results for read and write performance at different supply voltages.

Additionally, we also compared the post-layout simulation result with newly published state-of-the-art SRAM designs in Table 4. This table shows the comparison of the effective cost in terms of the performance and stability of the SRAM chip with the other existing SRAM design. The proposed chip shows a better cost than existing designs in terms of most of the parameters [20,28]. Here, we compared the proposed design with the best suitable 12T and recently published 12T SRAM designs. According to the published results RHPD-12T gives a better write performance than the proposed designs because it was designed for an improved write performance. We can also increase the write performance of the proposed SRAM design by adopting the technique from RHPD-12T [28]. Another proposed SRAM shows a better stability and read performance than RHPD-12. The proposed SRAM shows a similar stability than standard bit-interleaving and write assist 12T [20]. Moreover, our SRAM design has a 55 times greater read performance than bit-interleaving 12T. Hence, the proposed 12 SRAM will be a better choice under this category for future designs. We could not find a similar design for the power dissipation calculation; hence, we simply calculated the power dissipation and will try to compare it in future designs.

**Table 4.** Comparison of effective cost in terms of stability and performance.

| SRAM | RNM (mV) | SNM (mV) | Read (ps) | Write (ps) |
|---|---|---|---|---|
| RS12T | 343.4 | 347 | 76.1 | 73.9 |
| RSWAT12T | 317.8 | 347 | 126 | 46.7 |
| RSWAT12T2 | 317.8 | 347 | 121 | 44.9 |
| 12T [20] | 337.1 | 337.1 | 6970 | 92.1 |
| 12T [28] | 225 | 450 | 78.48 | 39.26 |
| 6T | 187.5 | 361.1 | - | - |

*4.3. Power Dissipation*

The physical length of the semiconductor device is decreased in order to achieve the required performance. However, at the same time, the power dissipation increases. The power dissipation increases in modern semiconductor devices due to short and narrow channel effects. However, at the same time, the portable device required a low-power memory. Static and dynamic (read and write operations) power dissipation is calculated using a post-layout netlist. A butterfly-based 512-byte array, memory controller, driver, decoder, and all required logic circuits are operating on a single supply. Hence, we calculated the total power dissipation by a single supply. However, this is not the sum of the clock signal because the clock input

is provided externally. We reduce the power dissipation using the virtual ground technique in the proposed SRAM. The virtual ground signal disables in the write and standby mode of operation. The dynamic power dissipation in the read and write mode of operations for the 512 byte SRAM cell is shown in Figure 29. We can save the dynamic power at scaled supply voltages, but, simultaneously, the read and write performance decreases. The simulation results are shown in Figure 28. We save 12.1% more dynamic power than the state-of-the-art design. The standby power of the proposed SRAM is also shown in Figure 30.

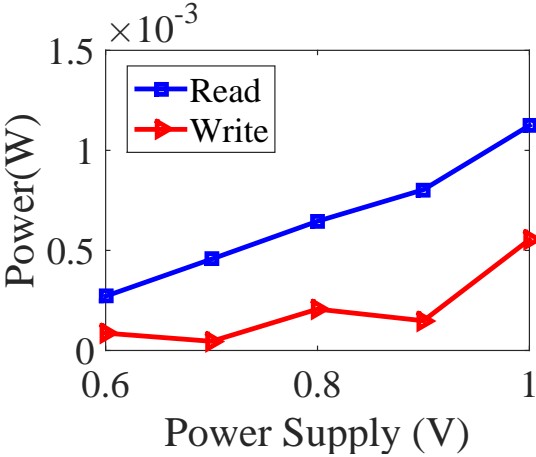

**Figure 29.** Dynamic power analysis for different supply voltages.

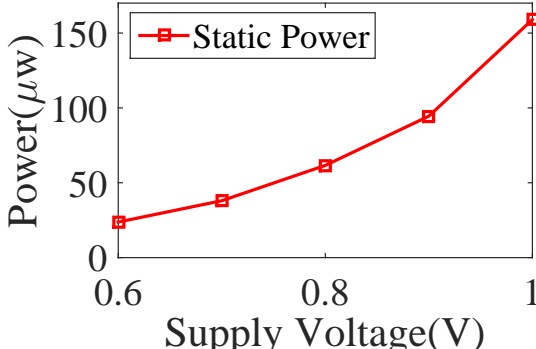

**Figure 30.** Static power analysis for different supply voltages.

## 5. Conclusions

In this paper, we have proposed three positive feedback-based SRAM bit-cells for highly reliable terrestrial applications. We have targeted a built-in computational memory-based microprocessor design. Hence, we have developed an isolated read port-based SRAM that can directly be used in artificial intelligence applications. The targeted applications are the convolutional neural network and other deep-learning-based microprocessor designs. The RSWA12T SRAM cell is useful for most challenging reliability situations. Compared with most of the considered SRAM cells, the proposed RSWA12T (second) SRAM has a comparable or better RNM and SNM at the nominal supply voltage. This SRAM also tolerates the process variation and other readability effects. Besides, the RSWA12T has a similar or lower area overhead compared with most of the considered state-of-the-art 12T tolerant SRAM bit-cells. Hence, the RSWA12T is an excellent choice for highly reliable high-density terrestrial applications at the nominal supply voltage. The write performance of the proposed SRAM cell is 80.4% greater than the existing 12T SRAM cell, whereas the read performance is 55.5% greater than the existing SRAM cell. The read stability of the proposed SRAM is 49.9% greater than the state-of-the-art 6T SRAM. The proposed SRAM cell and architecture also save more than 12% more power than the existing design.

**Author Contributions:** N.Y.: proposed idea, simulation, measurement, data analysis, and manuscript writing; Y.K.: simulation and manuscript writing; S.L.: simulation and manuscript writing; K.K.C.: project funding and supervisor. All authors have read and agreed to the published version of the manuscript.

**Funding:** This work is supported by the Industrial Core Technology Development Program of MOTIE/KEIT, KOREA. (#10083639, Development of Camera-based Real-time Artificial Intelligence System for Detecting Driving Environment & Recognizing Objects on Road Simultaneously).

**Institutional Review Board Statement:** Not applicable.

**Informed Consent Statement:** Not applicable.

**Acknowledgments:** We thank our colleagues from KETI and KEIT who provided insight and expertise that greatly assisted the research and greatly improved the manuscript.

**Conflicts of Interest:** The authors declare no conflict of interest.

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
