# Peer review of "Stable, Low Power and Bit-Interleaving Aware SRAM Memory for Multi-Core Processing Elements"

_electronics, doi:10.3390/electronics10212724_

Round 1

Reviewer 1 Report

This paper introduces a new SRAM design that improves the performance, power efficiency, and stability as compared to the existing SRAM. Generally, I think the overall flow of the paper is good, that authors discuss enough details of their designs. However, I have some major concerns about the current version.

The biggest concern I have is that the "state-of-the-art" baseline seems old (citation [18] published in 2014). It is not clear why the authors choose this architecture as the baseline. Without the comparison with some newer architectures (e.g., performance/area/power/stability comparison with other 12T SRAMs), it is hard to recognize the benefits of the proposed design in the current context. Furthermore,  it would be better to compare with state-of-the-art non-12T SRAM cells. 

Second, the paper lacks the overall system results in the context of computational memory applications. Although it is good to have better read/write performance, it is unclear how such benefits would reflect on the overall system performance when running a real application (e.g., CNN). The title mentioned this SRAM memory is designed for multi-core processing elements, it is unclear to me how the multi-core processing elements will use the proposed design to accelerate target applications. An extra introduction and results for the real use case in multi-core processing elements would be very beneficial to improve the paper.

The writing can also be improved in the current version. There are several typos and grammatical errors throughout the paper. For example, the abstract has the sentence - "We improve the write performance
11 by 28.15% compare with the state-of-the-art 12T SRAM design".

If a revision is necessary, I suggest the authors to:

  1. Add comparison experiments to more recent 12T SRAM
  2. Add comparison or discussion about other non-12T/6T SRAM
  3. Add introduction and experiments to multi-core processing elements with computational SRAM for a real application (e.g., DNN)

Author Response

Dear Reviewer
Greetings

We would like to thank you for the valuable comments. It will really improve our manuscript. Comments will also help in our future research work. We tried to give a response to your all comments. Please find the author responses report attached herewith. 

Regards
Nandakishor Yadav

Reviewer 2 Report

The authors developed an isolated read port-based SRAM 345 and focused its significance on computation-intensive artificial intelligence applications.

The paper is well motivated. It attacks a very relevant problem for artificial intelligence applications.

There is not enough information to understand how the experiments are performed. How the percentage of power reduction and performance are calculated was not clearly described

Section I and II are well written, but the figure descriptions can be improved. The authors can explain better how the 12T SRAM cell built-in computation 38 memories, can be beneficial for CNN architecture.  Such an explanation could make this paper more convincing and interesting. Here are some typos:

Page 1-typo in the table (‘Reed)

Page 13-typo in the graph (change in %)

Overall, I like the paper. But authors must need to address the above comments.

Author Response

(The authors gave the same response as above.)

Reviewer 3 Report

This paper mainly proposed a self-adaptive 12T SRAM cell to increase the read stability for multi-port operation. However, there are some comments could be listed as follows:

  1. The motivation of this work is not that clear. It is hard to link the proposed SRAM cell with CNN and CIM (as presented in the Introduction section). The proposed 12T SRAM doesn't support any computation operations. The motivation part should be modified.
  2.  In Figure 5, Figure 6, Figure 19 and Figure 20, the simulated PVT conditions of these simulated results should be added.
  3. A detailed comparison table should be added in this paper.
  4. In Figure 14, the layout of conventional 6T SRAM cell is not correct. Please refer to the compact cell design. The area overhead of the proposed cell, as shown in Fig 13, can be quite high.

Author Response

(The authors gave the same response as above.)

Round 2

Reviewer 1 Report

Authors add new results and discussion which address some of my concerns, especially the new result for a more recent SRAM design in Table 4. However, authors have not included sufficient illustration to interpret the results in Table 4. I suggest the authors include a more detailed illustration to emphasize how the Table 4 shows the proposed design is better than existing works.

Author Response

Dear Reviewer 

Greetings

Thank you very much for the feedback and your time. Please find the attached response file for the comments.

with kind regards

Nandakishor Yadav
